# Outdoor cycling activity affected by COVID-19 related epidemic-control-decisions

**Anne-Maria Schweizer**[1]*, **Anna Leiderer**[1], **Veronika Mitterwallner**[1], **Anna Walentowitz**[1], **Gregor Hans Mathes**[1], **Manuel Jonas Steinbauer**[1,2]

**1** Sport Ecology, Bayreuth Center of Ecology and Environmental Research (BayCEER) & Department of Sport Science, University of Bayreuth, Bayreuth, Bavaria, Germany, **2** Department of Biological Sciences, University of Bergen, Bergen, Norway

* Anne-Maria.Schweizer@uni-bayreuth.de

**Data Availability Statement:** The data underlying the results presented in the study are available at https://figshare.com/articles/software/Outdoor_cycling_activity_affected_by_COVID-19_related_

## Abstract

### Aim

The lockdown of sports infrastructure due to the COVID-19 pandemic has substantially shifted people's physical activity towards public green spaces. With Germany's lockdown as one of the more severe governmentally imposed epidemic-control-decisions, we tested to what extent the frequency of outdoor cycling activities changed from March to June 2020.

### Methods

User behaviour and frequency in 15 urban and 7 rural German public green spaces was quantified using cycling data from the fitness application Strava. Changes in cycling activities were analysed with four different generalised linear models, correcting for factors like weather conditions and temporal changes in the user base of the fitness application.

### Results

We found a clear increase in outdoor cycling sport activities in urban public green spaces in response to epidemic-control decisions (e.g. increase by 81% in April relative to the expected value (95% CI [48%, 110%])). In contrast, biking in rural areas showed no significant change with epidemic-control-decisions in place.

### Conclusion

Fitness App data, e.g. from Strava, can be used to monitor visitor behaviour and frequency. The increase in outdoor cycling activities during epidemic control decisions likely reflects a shift of sport activities from indoor and team sports to outdoor and individual sports. This highlights the importance of accessible green space for maintaining physical fitness and health. Beyond this shift, it is likely that outdoor activities may be of particularly importance for stress relief in times of crisis such as the current COVID-19 pandemic.

epidemic-control-decisions/13285808, doi: https://doi.org/10.6084/m9.figshare.13285808.v1.

**Funding:** This publication was funded by the German Research Foundation (DFG) and the University of Bayreuth in the funding programme Open Access Publishing.

**Competing interests:** The authors have declared that no competing interests exist.

## Introduction

In many countries of the world, governments issued epidemic-control-decisions at the beginning of 2020 to prevent the spreading and transmission of the highly contagious virus SARS-CoV-2. This novel virus causes COVID-19, a respiratory disease that can cause a variety of health issues [1] and can lead to death. The measures taken differed between countries and, in countries like Germany, even between federal states but with the joined aim of slowing down the spread of the virus. The results were immense constraints for the private life of citizens, leading to extensive changes in mobility, purchasing behaviour and environmental impact of whole populations [2, 3]. Facing rising case counts, the German Government issued progressively stricter policies. From March 12[th] 2020 on, events with an expected attendance of 1,000 people were prohibited [4] and since March 16[th], recreational facilities were closed to the public [5], including all tourism related facilities. On March 22[nd,] social distancing was enacted. The lockdown of sports infrastructure has substantially reduced the possibilities for leisure physical activity and shifted people's physical activity outdoors towards public green spaces. An increase of outdoor recreational activities by almost 300% during mobility confinements, compared to a 3-year average, was shown in Oslo, Norway [6]. In Germany, similar effects are indicated by search query data from Google Trends (Google Trends 2020) in terms of outdoor recreation key words like "Fahrrad" (bicycle) or "Wandern" (hiking). For those terms, search query in May 2020 were 40% higher than the average of the last 3 years. Trail remoteness and closed canopy cover were identified as important traits for trail selection during the crisis [6], suggesting that public green spaces facilitated social distancing and indirectly mitigated the spread of SARS-CoV-2 virus.

Understanding these changes in spatio-temporal patterns of physical activity is important, as there are strong indications that outdoor activities in public green spaces like forests and parks are a major strategy for coping with COVID-19 and the associated epidemic-control decisions [6–8]. With group sport activities prohibited, people may feel restricted to sedentary indoor activities, resulting in a potential reduction in physical exercise and increased sitting associated with a higher risk of chronic diseases or worsening physical constitution [9–13]. As preventive measure, it is recommended to exercise during lockdown, either at home or outdoors [14]. The risk of contracting COVID-19 does not decrease when exercising, but a healthy and fit body may handle an infection more successfully [15]. Recreational outdoor activities have been proven beneficial not only for physical but also for mental health [16], whereas green environments are more important for mental health benefits than urban grey environments [17]. Increased greenness is associated with stress reduction [18], lower likelihood of psychological distress and other positive mental health outcomes, especially with physical activity as a mediator for stress reduction [19, 20]. This indicates that access to public green spaces also enhances the resilience of individuals to cope with crises like a pandemic [8].

Over the last years, physical activity is increasingly supported by various fitness applications (short: fitness apps) and the user base of such is growing rapidly [21]. With the help of a smartphone or other GPS devices, anyone can track their geographical data and upload it to online platforms like social media or fitness apps. These voluntarily offered data have great potential as useful management tools by providing spatio-temporal information [e.g. 22]. An app can act as an indicator for visitor figures in nature parks and urban green areas [23–27] and thus provide solid data for quantifying frequencies of outdoor sport activities such as cycling, running, or hiking.

In this study, we quantify the effects of governmental epidemic-control-decisions on outdoor cycling activities in Germany. For this, we use publicly available data of cycling behaviour from Strava [28], a popular fitness tracking app. Germany's regulations for social distancing

got progressively stricter since the beginning of March 2020 [5], although measures, and their moment of implementation, differed between federal states. With indoor sport facilities closed and group sports prohibited, we expect an immediate increase in outdoor cycling activities with the implementation of epidemic-control-decisions. We further test if observed changes differ between public green spaces in densely populated areas (major cities), and less accessible, but greener countryside locations (nature parks). Results can support the use of data from fitness apps for management purposes.

## Materials and methods

### General procedure

Systematic changes in the frequency of outdoor cycling activities were analysed using publicly available data from the fitness app Strava [28], which aims to support sportspeople during physical activity. The GPS-trail of users is broken down by Strava into snippets, called segments, oftentimes describing one specific activity like climbing or descending. The Strava app was first released in 2009 and by now has built up a substantial user base. Users of the Strava fitness app can compare their latest performance with their past performances and the performance of other users. Globally, user numbers are constantly rising with 8.2 million in March 2015 [29] and reaching up to 42 million in July 2019 [30]. The current accession rate is 1 million new users every 30 days [31]. When using this data to quantify changes in spatiotemporal patterns of sport activities, one needs to control for the constantly growing user base and consider possible influential factors like weather conditions. For each analysed segment, we thus aggregated available cycling data per month and quantified their dependence on monthly sunshine hours as well as their increase with time for the respective months. Resulting models were used to predict the expected cycling activities for the respective segment without a lockdown situation and were subsequently compared to the recorded values. We excluded months of winter due to infrequent sports activity and truncated the data to start in 2012, as the app was used too infrequently beforehand to support reliable analyses.

### Study areas and sample period

Germany is undergoing a long-term process of urbanization, resulting in 30% of the German population living in large cities by 2015 [32]. We chose to conduct this study using data from the 15 largest German cities in terms of population and 7 randomly chosen German nature parks to contrast user frequencies of urban and rural public green spaces (see Tables 1 and 2 for additional information). Although most epidemic-control-decisions were issued in March 2020, we consider April, May, and June 2020 to be most influenced by them, due to people adjusting to the new policies. Springtime starts with March and April in Germany, which was marked in 2020 by persisting fair weather [33].

**Table 1. Overview of sampled nature parks.** Area figures as reported on the website Naturparke.de [34].

| Nature park | Federate state | Area [km2] (2020) |
|---|---|---|
| Bavarian Forest | Bavaria | 2780 |
| Black Forest | Baden-Wurttemberg | 3750 |
| Harz (Lower Saxony) | Lower Saxony | 2810 |
| Hohe Mark | North Rhine-Westphalia | 1978 |
| Lüneburger Heath | Lower Saxony | 1070 |
| Teutoburg Forest/Egge Hills | North Rhine-Westphalia | 2736 |
| Usedom Island | Mecklenburg-Western Pomerania | 632 |

**Table 2. Overview of sampled cities.** Area and population figures as reported by the Federal Bureau of Statistics [35], with area figures rounded mathematically.

| City | Federate state | Area [km2] (2016) | Population (2019) |
|---|---|---|---|
| Berlin | Berlin | 894 | 3 669 491 |
| Bremen | Bremen | 524 | 567 559 |
| Cologne | North Rhine-Westphalia | 405 | 1 087 863 |
| Dortmund | North Rhine-Westphalia | 281 | 588 250 |
| Dresden | Saxony | 328 | 556 058 |
| Duisburg | North Rhine-Westphalia | 233 | 498 686 |
| Dusseldorf | North Rhine-Westphalia | 217 | 621 877 |
| Essen | Lower Saxony | 210 | 582 760 |
| Frankfurt am Main | Hesse | 248 | 763 380 |
| Hamburg | Hamburg | 755 | 1 847 253 |
| Hannover | Lower Saxony | 204 | 536 925 |
| Leipzig | Saxony | 298 | 593 145 |
| Munich | Bavaria | 311 | 1 484 226 |
| Nuremberg | Bavaria | 186 | 518, 70 |
| Stuttgart | Baden-Wurttemberg | 207 | 635 911 |

## Data acquisition

We only analysed segments registering the highest user numbers per focal area, as suggested by Norman & Pickering [22]. In Germany, the most popular individual sport activities are cycling and running [36]. Since the data on running segments in rural areas and nature parks is too sparse to be used, we concentrated our efforts on cycling segments. For each sample city, we chose and downloaded two cycling segments, resulting in 30 examined urban segments. We only selected segments with high user frequency with more than 1500 attempts and more than 350 users reporting an attempt. Other requirements were for the segment to pass alongside or cross a body of water or park to qualify it as a green space, to have a minimum length of 500 meters and a comparable mean count of attempts per user. This way, we excluded commuting activities from our analysis. Furthermore, we excluded segments used for any kind of event, e.g. a city run. The seven focal rural areas were selected at random from all 104 German nature parks [34]. For each rural area, two cycling segments were chosen, meeting the same requirements as for segments in urban areas, although requirements concerning user frequency had to be reduced to include segments with more than 600 attempts and more than 200 users reporting an attempt. User frequency in nature parks was generally lower due to their distance to highly populated areas and their overall bigger total area. We downloaded data from Strava [28] in June and July 2020, and cleaned data by excluding duplicates, resulting in 3499 data points overall. To correct for the effect that sunshine hours influence individual's exercising behaviour [37, 38], we identified the closest weather station to focal segments from the German weather service DWD (Deutscher Wetterdienst) and aggregated the number of sunshine hours per month for the period investigated.

## Data analysis

We implemented generalised linear models with Poisson family error to quantify the general increase in cycling activities. Models were trained individually for each segment and month, based on data from preceding years without COVID-19 influence, incorporating the continuously growing user base of the fitness application, as well as accounting for the effect of changing weather conditions. Models were implemented with four alternative settings: (A) only

including the development of occurrences of sport activities with time [cycling activity ~ year], (B) adding the effect of sunshine hours [cycling activity ~ year + sunshine hours] and (C) additionally including the interaction between sunshine hours and time [cycling activity ~ year + sunshine hours + year: sunshine hours]. A (D) Null Model was also implemented. We discerned the best fitting model using Akaike's Information Criterion [39].

Since the data was not normally distributed, the user frequency, as predicted by the model without COVID-19 influence, was compared to the actual user frequencies extracted from Strava [28] via non-parametric bootstrapping. 95% Confidence Intervals were then calculated using the adjusted bootstrap percentile (BCa) method [40]. This approach was implemented for the months with COVID-19 influence (March–June 2020), to quantify the effect of epidemic-control-decisions as well as for other months (July–October 2019). That way, the approach was validated to predict realistic values when COVID-19 and epidemic-control-decisions were not present. This means the difference between expected and measured activities for March—June 2020 can be attributed to COVID-19 related changes.

We used R version 3.6.2 for all data analysis, including the packages boot [41, 42], flextable [43], here [44], lubridate [45], MuMIn [46], officer [47], readxl [48] and tidyverse [49]. For creating the map in Fig 1 we used ciTools [50], gridBase [51], Lattice [52] and raster [53].

## Results

The best fitting model was discerned to be Model A, using Akaike's Information Criterion. Henceforth we will focus primarily on Model A, which describes the development of occurrences of sport activities over time, and performs best in 69% of cases. With epidemic-control-decisions in place (March–June 2020), we see a significant increase of 55% (95% CI [45%, 74%]) cycling activity in urban public green spaces. In contrast, rural public green spaces show no significant differences. During times without the influence of epidemic-control-decisions (July–October 2019), all three models show either no significant cycling frequency or slightly overestimate cycling frequency in urban and rural public green spaces (see Fig 2). Our models show a significant increase of up to 81% in cycling activity in urban public green spaces (e.g. Model A, April: med. diff. = 54.4, percent diff. = 81%, 95% CI [48%, 110%]). For a detailed visualization of every sampling site see Fig 1.

## Discussion

In this study, we showed that epidemic-control-decisions in Germany in response to the COVID-19 pandemic have caused a rise in cycling activity in urban areas, reflected by user frequencies of the Strava fitness app [28]. Our findings align with augmented sales of bicycles and biking equipment during the COVID-19 pandemic, going up 9.2% in Germany alone [55]. The effects differed between urban and rural public green spaces, whereby rural areas registered no significant change in user frequency with epidemic-control-decisions in place (March–June 2020) (see Fig 2A–2C). Urban areas, however, show a general increase of 55% in cycling activity, with a maximum increase of 81% per month. Similar effects were found in Oslo, Norway, where outdoor recreational activity during mobility confinement increased by 300% [6]. The Google Mobility Report of May 2020 for Germany [56] states that visitor frequency for public green spaces, including national parks, public beaches, marinas, dog parks, plazas and public gardens, have gone up by 225%. It is to be expected that easy to access public green spaces designed for tourism, including beaches and public gardens, will have higher activity fluctuations than regular public green spaces, which were used in this study. Although our numbers turn out lower compared to similar studies, we observe the same effect.

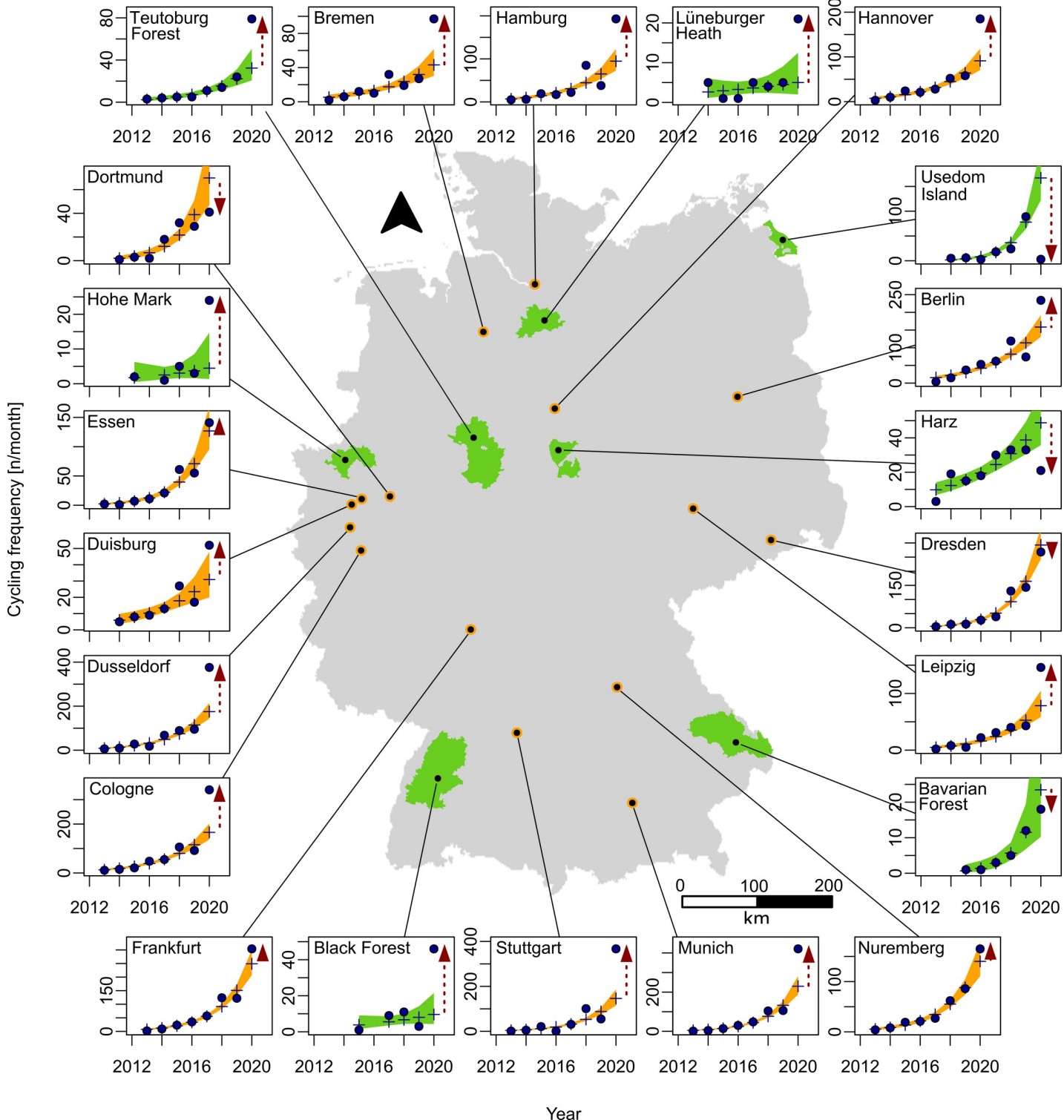

**Fig 1. Map of study sites and respective cycling frequency.** Cycling frequency displayed as measured (point) and predicted (plus, Modell A with user numbers as factor) values for the month April for all urban (orange) and rural (green) sample sites. The red dashed arrows indicates a Covid-19 related difference and the direction of change in user frequency. The background shows the outlines of Germany with the locations of focal cities and nature parks [54].

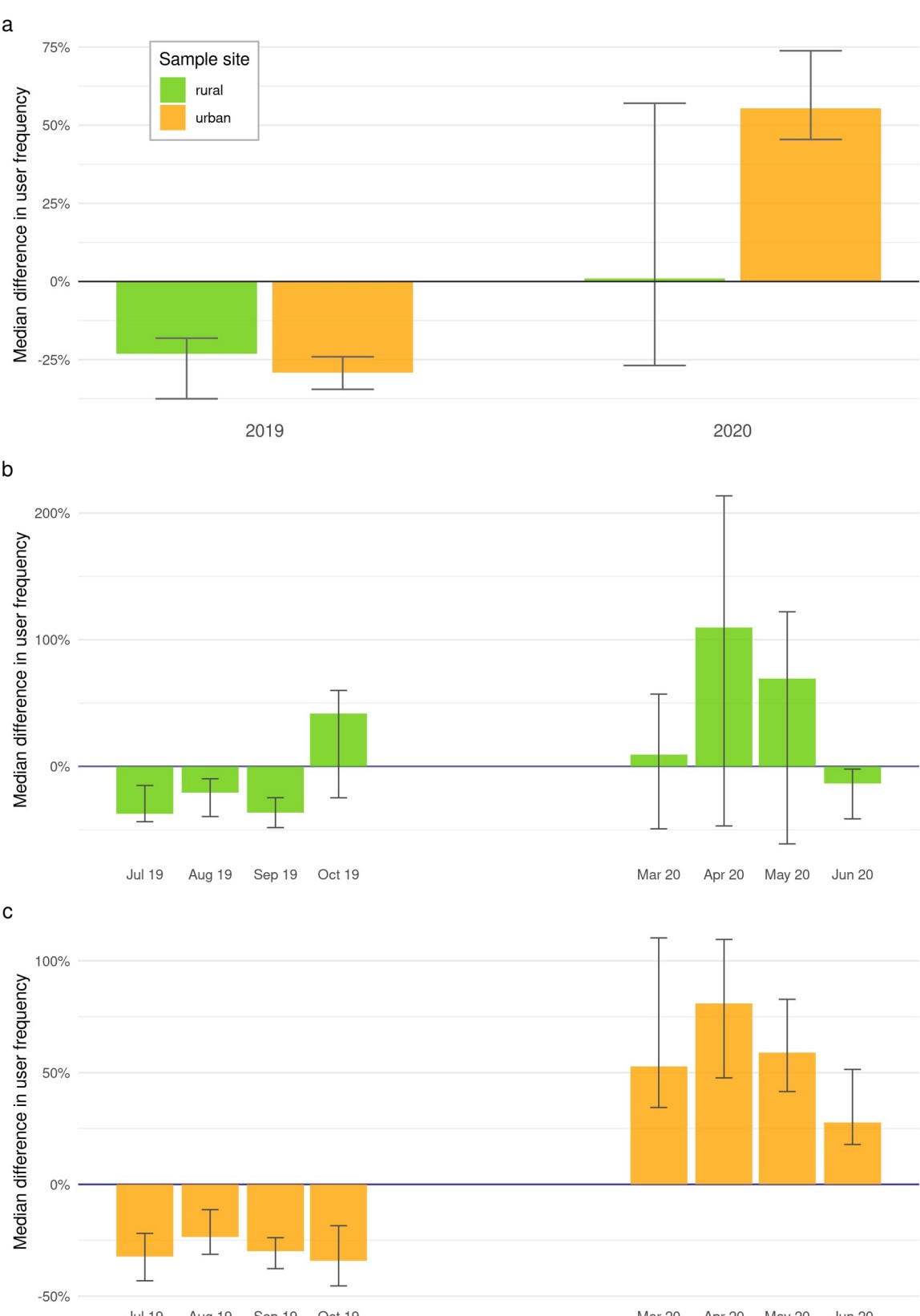

**Fig 2. Comparison of urban and rural cycling behaviour.** Graphic a displays a summary of Model A [cycling activity ~ year]. The plot shows the median differences in user frequency, detected in rural (green) and urban (yellow) public green spaces in times with influence of epidemic-control-decisions (2020) and without (2019). Bars indicate mean difference, whiskers show 95% Confidence Intervals. Graphic b and c display median differences in user frequency, calculated by Modell A [cycling activity ~ year]. Graphic b shows results for rural and Graphic c for urban public green spaces in months with influence of epidemic-control-decisions (March–June 2020) and without (July–October 2019).

Looking at other studies, there is evidence of increased sitting due to governmental epidemic-control-decisions [11, 13]. However, the effects upon physical activity are not uniform, with some individuals decreasing their physical activity while others increased their activity. It is possible that those who increased their physical activity were individuals who successfully transitioned from gyms and recreation centers to urban green spaces.

The increase in outdoor activities is a direct result of the prohibition of team sports and closed sports infrastructure, which prevent most indoor sport activities. It may also reflect the larger amount of available free time, resulting from the rising count of people working reduced hours [57]. Their number increased dramatically by over seventeen thousand percent (17 447%) from 43 000 affected people in January to 7 502 265 in April [58]. We suspect that the rise in user frequency in urban public green spaces observed in this study is pronounced, in comparison with rural public green spaces, because of high urban population density and less available green space per individual. While a shift to outdoor sport may be important to maintain physical activity with epidemic-control decisions in place, outdoor sports also facilitate psychological welfare [18], which is of similar importance for a population, especially during a pandemic such as COVID-19. The COVID-19 pandemic, and the resulting epidemic-control-decisions, have added stressors to people's daily life. The initial lack of information about the novel SARS-CoV-2 virus, followed by a plethora of different epidemic-control-decisions, might have unsettled the public. The closing of educational facilities for a limited time resulted in teaching arrangements varying majorly between facilities. Legal guardians and students alike had trouble to adapt to these drastically different teaching arrangements, mostly because of lacking communication with school staff [59]. According to a non-representative study [60], most legal guardians were facing a huge dual burden of working from home and supervising their offspring's home-schooling, let alone minding household chores. Social distancing and the temporary closing of businesses and facilities caused people to be restricted to their own home. The closing of sports facilities implies that all 27.6 million members of German sport clubs [61] are no longer able to exercise in the weekly training sessions. Furthermore, the 11.7 million members of the German gyms [62] are forced to either skip their training or find new ways of being physically active. At times, people were only allowed outside alone or in small groups when shopping for essentials or when exercising. Social gatherings of all kinds had to be cancelled, leading to a disrupted and reduced social life. These policies also led to a surge in domestic violence [63, 64], although its extend will only become clear with next year's criminal statistics. Lastly, people might have felt afraid for themselves or their loved ones contracting COVID-19, have had to deal with being infected or with bereavement. The demand for mental health consultations has been on the rise [65], with counsellors trying to relief patient's worries and stress. All these stressors of people being restricted to their own homes, coping with completely new challenges concerning employment, care work, domestic work and social life, had to be compensated. Our data, along with Venter et al. [6], suggests that for stress relief, people turned to individual outdoor sport activities like cycling, using available public green spaces at a higher rate than before the COVID-19 pandemic.

## Consequences for city planning

Since a surge in outdoor activities was only observed in urban public green spaces, they seem to be more crucial for short-term stress relief than rural public green spaces, presumably due to high population density in cities. A recent study [66] found an increase in public green spaces in major cities by 3.2% since 1996. Currently, public green spaces make up 10.9% of major German cities, which is above the national average of 6%. Although less urbanized and rural areas have less public green spaces inside city borders on a percental basis, green spaces situated outside of city limits are much more accessible. These vast rural and natural green spaces lead to a much more relaxed resident-to-green-space ratio in rural areas compared to urban areas. Additionally, epidemic-control-decisions have led to a steep decline in national and international tourism [5] and in turn to lower visitor frequencies of nature parks. Advantages of green spaces are numerous. While promoting mental and physical health [16], they also absorb $CO_2$, provide shade, buffer noise with greenery and provide refuge for wild species [67]. Since people seem to actively seek out green spaces for their stress relieving properties, these areas mitigate the spread of COVID-19 and promote social distancing. Future city planning should continue to establish numerous, accessible urban public green spaces to stabilize the mental and physical well-being of the population, to improve the populations resilience and to provide opportunity for stress relief.

## Methodological aspects and the potential of fitness applications

For our analysis, we build four models with different settings. Model A models the development of occurrences of sport activities over time. Model B added the effect of sunshine hours, a factor that is proven to influence outdoor activity [37, 38]. Model C included the interaction between sunshine hours and time. We also implemented a Null model (D). Surprisingly, the best fitting model was discerned to be Model A instead of any other model containing the influence of sunshine hours. Rising user numbers, new segments and distribution of active user on segments are currently not in equilibrium and dynamically changing. Thus, these factors explain more variation of the data than sunshine hours. This may change over time when user numbers develop at a steadier rate and users are more evenly distributed throughout urban and rural areas. Since the Strava app [28] does not focus on orientation, user numbers may stay low in rural areas compared to urban areas. It is important to orientate oneself, especially in rural areas where users are potentially alone and cannot ask for directions. To that end, users might utilise other fitness apps.

The results presented in this paper are based on the information provided by one fitness app and thus the changes of outdoor cycling activities during lockdown are only partly reflected, although Strava is one of the most popular and widespread fitness apps in Germany. Behavioural changes of people exercising outdoors without tracking their activities are not captured in our dataset. Furthermore, user shifts between fitness apps are not reflected as our results are only based on Strava data. Nevertheless, our analysis provides valuable insights into activity changes of cyclists with georeferenced user-generated open-access data. The used approach is suitable to assess rapid changes occurring during the sudden oncoming of the global COVID-19 pandemic, which did not allow for stratified planned monitoring.

Some studies have already conducted research with Strava data, although none of them have attempted to predict future developments of user frequencies [6, 22]. Monitoring user frequency of green spaces can be achieved with the help of fitness apps, as shown in this paper. Via observing current user frequencies and predicting future frequencies, tending plans for green spaces can be compiled. This method will prove very useful for management purposes of urban and rural green spaces.

Strava does not directly provide information concerning the gender distribution of users. However, from Strava's annual report of 2018 [68] it can be concluded from cycling specific upload figures, that the distribution is massively skewed, with most of the users identifying as male (81%). This is in high contrast to the 51% men's quota in recreational cycling in Germany [69]. Since Strava's main feature is its high score table, the app itself creates a highly competitive setting. In most societies, especially western ones, people identifying as male tend to be more positive about and to seek out competition than people identifying as female [70, 71], which could explain the uneven gender distribution on the fitness app. Hence, our findings apply to people identifying as male and not to the general public.

## Conclusion

Data from fitness apps can indicate user behaviour and frequency in green spaces, as shown using the Strava app. Therefore, fitness apps can be a useful management tool for assessing visitor frequency. In this study, we found evidence that outdoor cycling sport activities increased in urban public green spaces in Germany in response to COVID-19 related epidemic-control decisions.

## Acknowledgments

We thank Charlotte Regina Müller and Sofie Paulus for support acquiring Strava Data. Furthermore, we thank Dr. Matthias Biber and Dr. Volker Audorff for comments on the manuscript.

## Author Contributions

**Conceptualization:** Anne-Maria Schweizer, Anna Leiderer, Anna Walentowitz, Manuel Jonas Steinbauer.

**Data curation:** Anne-Maria Schweizer, Gregor Hans Mathes.

**Formal analysis:** Anne-Maria Schweizer, Gregor Hans Mathes, Manuel Jonas Steinbauer.

**Investigation:** Anna Leiderer.

**Methodology:** Manuel Jonas Steinbauer.

**Project administration:** Anne-Maria Schweizer.

**Software:** Manuel Jonas Steinbauer.

**Supervision:** Manuel Jonas Steinbauer.

**Validation:** Anne-Maria Schweizer.

**Visualization:** Anna Walentowitz.

**Writing – original draft:** Anne-Maria Schweizer.

**Writing – review & editing:** Anne-Maria Schweizer, Anna Leiderer, Veronika Mitterwallner, Anna Walentowitz, Gregor Hans Mathes, Manuel Jonas Steinbauer.

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
