## [Decision Letter · Decision Letter 0]

24 Feb 2021

PONE-D-21-00324

Outdoor cycling activity affected by COVID-19 related epidemic-control-decisions

PLOS ONE

Dear Dr. Schweizer,

Thank you for submitting your manuscript to PLOS ONE. After careful consideration, we feel that it has merit but does not fully meet PLOS ONE’s publication criteria as it currently stands. Therefore, we invite you to submit a revised version of the manuscript that addresses the points raised during the review process.

We look forward to receiving your revised manuscript.

Kind regards,

Pasquale Avino, Ph.D.

Academic Editor

PLOS ONE

Journal Requirements:

Reviewers' comments:

Reviewer #1: This is an excellent project with useful novel information. The manuscript is well executed though I do have some suggestions for minor revisions below. My one significant suggestion is to incorporate some of the literate reporting how the pandemic may have affected physical activity and sedentary behavior in general. See my comments below.

Introduction

“The results were immense constraints for the private life of citizen…” pluralize citizen.

“Hence, in these days of the crisis, it seemed like people were searching for physical and emotional compensation in public green spaces” This sentence is confusing. You have provided evidence that people may have been seeking green space for physical activity. You have not made a case for this being done for emotional reasons. I don’t doubt that there may have been a psychological component to the increased use of green spaces but you have not made that point.

“With group sport activities prohibited, people may feel restricted to sedentary indoor activities, resulting in a lack of physical exercise associated with a higher risk of chronic diseases or worsening physical constitution” There is emerging evidence supporting the idea that the pandemic disrupted physical activity and increased sitting. I would encourage the following revision and cite the following papers “With group sport activities prohibited, people may feel restricted to sedentary indoor activities, resulting in a potential reductions in physical exercise and increased sitting associated with a higher risk of chronic diseases or worsening physical constitution”

• Barkley, J.E., A. Lepp, E. Glickman, G. Farnell, J. Beiting, R. Wiet, and B. Dowdell (2020) The Acute Effects of the COVID-19 Pandemic on Physical Activity and Sedentary Behavior in University Students and Employees. International Journal of Exercise Science. 13(5): 1326-1339

• (1, 29, 33, 42, 43

• Ammar A, et al. On behalf of the eclb-covid19 consortium. Effects of covid-19 home confinement on eating behaviour and physical activity: Results of the eclb-covid19 international online survey. Nutrients 12(6):1583, 2020.

• López-Bueno R, Calatayud J, Andersen LL, Balsalobre-Fernández C, Casaña J, Casajús JA, Smith L, LópezSánchez GF. Immediate impact of the covid-19 confinement on physical activity levels in spanish adults. Sustainability 12(14):1-10, 2020

• Meyer J, McDowell C, Lansing J, Brower C, Smith L, Tully M, Herring M. Changes in physical activity and sedentary behaviour due to the covid-19 outbreak and associations with mental health in 3,052 us adults. Cambridge Open Engage 2020

Discussion

“It may also reflect the larger amount of free time available to many people, resulting from reduced working hours [53], which increased…” I believe you mean “working hours which decreased…”

This section is well done. The only other suggestion I have is to incorporate some of the previous research examining the impact of the COVID pandemic upon physical activity. In the studies below there is evidence of increased sitting. However, the effects upon physical activity are not uniform. Some individuals decreased PA while others increased. Your findings may offer a mechanism through which this effect occurred. It may be those that increased PA were the same types of individuals who successfully transitioned from gyms and recreation centers to urban green spaces. I would encourage you to briefly incorporate this notion into your discussion.

• Barkley, J.E., A. Lepp, E. Glickman, G. Farnell, J. Beiting, R. Wiet, and B. Dowdell (2020) The Acute Effects of the COVID-19 Pandemic on Physical Activity and Sedentary Behavior in University Students and Employees. International Journal of Exercise Science. 13(5): 1326-1339

• Meyer J, McDowell C, Lansing J, Brower C, Smith L, Tully M, Herring M. Changes in physical activity and sedentary behaviour due to the covid-19 outbreak and associations with mental health in 3,052 us adults. Cambridge Open Engage 2020

---

## [Author Response · Author response to Decision Letter 0]

10 Mar 2021

Response to Reviewers

Re-submission Date: 10th March 2021

Dear Reviewer,

We are delighted by your swift response and helpful critique and we greatly appreciate the feedback. In the following, we listed your suggestions and our adjustments to accommodate your input.

1. Suggestion: “The results were immense constraints for the private life of citizen…” pluralize citizen.

Adjustment: The mistake was corrected. (Lines 45)

2. Suggestion: “Hence, in these days of the crisis, it seemed like people were searching for physical and emotional compensation in public green spaces” This sentence is confusing. You have provided evidence that people may have been seeking green space for physical activity. You have not made a case for this being done for emotional reasons. I don’t doubt that there may have been a psychological component to the increased use of green spaces but you have not made that point.

Adjustment: We agreed and deleted the sentence in question. (Lines 59-60)

3. Suggestion: “With group sport activities prohibited, people may feel restricted to sedentary indoor activities, resulting in a lack of physical exercise associated with a higher risk of chronic diseases or worsening physical constitution” There is emerging evidence supporting the idea that the pandemic disrupted physical activity and increased sitting. I would encourage the following revision and cite the following papers “With group sport activities prohibited, people may feel restricted to sedentary indoor activities, resulting in a potential reductions in physical exercise and increased sitting associated with a higher risk of chronic diseases or worsening physical constitution”

• Barkley, J.E., A. Lepp, E. Glickman, G. Farnell, J. Beiting, R. Wiet, and B. Dowdell (2020) The Acute Effects of the COVID-19 Pandemic on Physical Activity and Sedentary Behavior in University Students and Employees. International Journal of Exercise Science. 13(5): 1326-1339

• Ammar A, et al. On behalf of the eclb-covid19 consortium. Effects of covid-19 home confinement on eating behaviour and physical activity: Results of the eclb-covid19 international online survey. Nutrients 12(6):1583, 2020.

• López-Bueno R, Calatayud J, Andersen LL, Balsalobre-Fernández C, Casaña J, Casajús JA, Smith L, LópezSánchez GF. Immediate impact of the covid-19 confinement on physical activity levels in spanish adults. Sustainability 12(14):1-10, 2020

• Meyer J, McDowell C, Lansing J, Brower C, Smith L, Tully M, Herring M. Changes in physical activity and sedentary behaviour due to the covid-19 outbreak and associations with mental health in 3,052 us adults. Cambridge Open Engage 2020

Adjustment: We agreed and incorporated the suggested sources into our manuscript. (Lines 64-65, see also References)

4. Suggestion: “It may also reflect the larger amount of free time available to many people, resulting from reduced working hours [53], which increased…” I believe you mean “working hours which decreased…”

Adjustment: The paragraph in question has been rewritten for more clarity. (Lines 218-219)

5. Suggestion: This section is well done [Discussion]. The only other suggestion I have is to incorporate some of the previous research examining the impact of the COVID pandemic upon physical activity. In the studies below there is evidence of increased sitting. However, the effects upon physical activity are not uniform. Some individuals decreased PA while others increased. Your findings may offer a mechanism through which this effect occurred. It may be those that increased PA were the same types of individuals who successfully transitioned from gyms and recreation centers to urban green spaces. I would encourage you to briefly incorporate this notion into your discussion.

• Barkley, J.E., A. Lepp, E. Glickman, G. Farnell, J. Beiting, R. Wiet, and B. Dowdell (2020) The Acute Effects of the COVID-19 Pandemic on Physical Activity and Sedentary Behavior in University Students and Employees. International Journal of Exercise Science. 13(5): 1326-1339

• Meyer J, McDowell C, Lansing J, Brower C, Smith L, Tully M, Herring M. Changes in physical activity and sedentary behaviour due to the covid-19 outbreak and associations with mental health in 3,052 us adults. Cambridge Open Engage 2020

Adjustment: We agreed wholeheartedly and composed a short new paragraph to talk about this. (Lines 211-215)

We hope to have incorporated all suggestions to your satisfaction. Thank you again for your helpful feedback.

---

## [Editor Report · Decision Letter 1]

16 Mar 2021

Outdoor cycling activity affected by COVID-19 related epidemic-control-decisions

PONE-D-21-00324R1

Dear Dr. Schweizer,

We’re pleased to inform you that your manuscript has been judged scientifically suitable for publication and will be formally accepted for publication once it meets all outstanding technical requirements.

Kind regards,

Pasquale Avino, Ph.D.

Academic Editor

PLOS ONE

---

## [Editor Report · Acceptance letter]

21 Apr 2021

PONE-D-21-00324R1 

Outdoor cycling activity affected by COVID-19 related epidemic-control-decisions 

Dear Dr. Schweizer:

I'm pleased to inform you that your manuscript has been deemed suitable for publication in PLOS ONE. Congratulations! Your manuscript is now with our production department. 

Kind regards, 

on behalf of

Professor Pasquale Avino 

Academic Editor

PLOS ONE